

# Information dynamics in a model with Hilbert space fragmentation

**Dominik Hahn, Paul A. McClarty and David J. Luitz**

Max Planck Institute for the Physics of Complex Systems,
Nöthnitzer Str. 38, 01187 Dresden, Germany

## Abstract

The fully frustrated ladder – a quasi-1D geometrically frustrated spin one half Heisenberg model – is non-integrable with local conserved quantities on rungs of the ladder, inducing the local fragmentation of the Hilbert space into sectors composed of singlets and triplets on rungs. We explore the far-from-equilibrium dynamics of this model through the entanglement entropy and out-of-time-ordered correlators (OTOC). The post-quench dynamics of the entanglement entropy is highly anomalous as it shows clear non-damped revivals that emerge from short connected chunks of triplets. We find that the maximum value of the entropy follows from a picture where coherences between different fragments co-exist with perfect thermalization within each fragment. This means that the eigenstate thermalization hypothesis holds within all sufficiently large Hilbert space fragments. The OTOC shows short distance oscillations arising from short coupled fragments, which become decoherent at longer distances, and a sub-ballistic spreading and long distance exponential decay stemming from an emergent length scale tied to fragmentation.

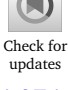
doi:10.21468/SciPostPhys.11.4.074

# 1  Introduction

The study of non-equilibrium quantum dynamics is one of the main frontiers of contemporary physics. While much remains to be explored and understood there are a few central results that help to frame questions of current interest. One of these is the observation that observables tend to thermalize at long times in local quantum many-body systems in the sense that the Gibbs ensemble of statistical mechanics becomes a good description of the asymptotic properties of the system even when the initial state is a single eigenstate. This commonly observed feature of most local quantum Hamiltonians has been elevated to a principle called the eigenstate thermalization hypothesis (ETH) [1–6]. Efforts to explore the validity of the ETH have unveiled a number of exceptions. Foremost among these are integrable models [7] that are found to relax to a generalized version of the Gibbs ensemble that includes, in addition to the energy, other conserved quantities that exist in such models [8–10]. Another well-known exception is the many-body localized (MBL) phase [11–18] in interacting strongly disordered systems (at least in one dimension) in which every eigenstate is localized and where local conserved quantities are emergent within the MBL phase [19,20].

More recently still, several models were found, where the many-body spectrum is sprinkled with highly athermal states that, by the nature of many-body spectra, live nearby in energy to thermal states in the middle of the spectrum [21–26, 26–37]. These unusual states are called many-body scars. Unlike MBL, many-body scars are present, by definition, in models without disorder and lead to a weak form of ergodicity breaking that is highly dependent on the initial state. A natural question is then whether there are disorder-free models where, similarly to the case of MBL, there is an exponentially large number of athermal states. This question has been answered in the affirmative through models with local conservation laws, local kinematics constraints, flat bands or geometric frustration [32,38–51].

In all these models the Hilbert space fragments into an exponentially large number of dynamically disconnected sectors with important consequences for observable properties. In particular, ETH is violated. For example, the spectra of these models feature predominantly athermal eigenstate entanglement entropies [41, 42]. Dynamical signatures of anomalous thermalization have been observed in various cases: including finite asymptotic autocorrelation functions [41], the presence of revivals [45], athermal plateaux in entanglement entropy following a quench [47] and finite asymptotic fidelities [47].

There are a number of natural questions that may be asked of all these models. How can we characterize the asymptotic, or long time, behavior of the model? To what extent does this depend on the initial state? What are the asymptotic states within dynamically disconnected fragments and how do these collectively lead to violations of ETH? Can we understand the dynamics of operator spreading in terms of the fragmentation picture? What is the effect of

weakly lifting the frustration thereby breaking the local conservation laws? In this paper, we systematically investigate features of the dynamics in one model with a fragmented Hilbert space: the fully frustrated ladder [43, 52–54]. This model, that we introduce in detail in the next section, is notable for the transparent nature of the fragmentation and is therefore highly suited to addressing all these questions in an intuitive way. In particular, one can readily see that the model is non-integrable, that the Hilbert space is fragmented and one can easily determine the nature of these fragments. A brief calculation enumerates all the fragments and reveals an emergent localization length that is a property of the full fragmented Hilbert space and, which is therefore, not restricted to low energies.

In the following we study the thermalization of subsectors and the interplay between ETH in large fragments and the violation of ETH in the gross features of the model. To do this we study the evolution of the entanglement entropy. We show that it exhibits recurrences and that the long time entanglement entropy evolution changes with the choice of initial state — both clear signs of athermal behavior. We also show that we get an excellent correspondence between the observed long-time properties and the entropy computed from states that are randomized within each conserved sector. This means that the system thermalizes as much as it can in the conventional sense of ETH but the small fragments overwhelm the gross behavior of the system so that it is highly athermal overall. The behavior of the small fragments is therefore the determining factor in the dynamics and we explore their effect on the dynamics in detail. In particular, we consider operator spreading of an operator $A(t)$ through the squared commutator $[A(t), B(0)]^2$ where $A$ and $B$ are local $S_z$ operators. The non-trivial part of this commutator is the out-of-time-ordered correlator (OTOC) that in thermalizing systems diagnoses the onset of chaos [55, 56]. We find that the spreading of local $S_z$ operators occurs sub-ballistically with an emergent length scale that can be captured within the fragmentation picture. The effect of local short chain fragments is reflected in short distance persistent oscillations in the OTOC whose features we capture semi-analytically. Finally, we study the lifting of frustration with a frustration-breaking parameter $\delta$ showing that it exhibits $\delta t$ scaling so that an arbitrary small $\delta$ leads to thermalizing behavior in the thermodynamic limit. In short, the model exhibits unusual thermalizing dynamics at odds with ETH that we can understand in detail from the fragmentation picture. While the model is interesting on its own, we discuss in the conclusions some features of the dynamics that are expected to occur in all models with Hilbert space fragmentation.

## 2 Model and Setup

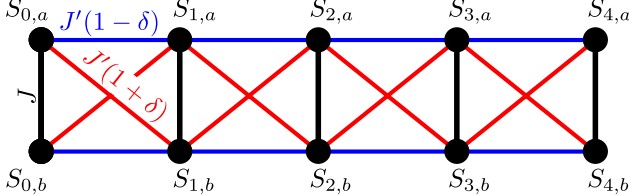

Figure 1: The frustrated spin $\frac{1}{2}$ Heisenberg ladder. The spins are located on the vertices. The couplings between different spins are given by $J$ (black coupling), $J'(1+\delta)$ (red) and $J'(1-\delta)$ (blue). For convenience we refer to one leg of the ladder as leg $a$, the other leg is $b$. The fully frustrated limit exhibiting Hilbert space fragmentation is for $\delta = 0$.

The frustrated Heisenberg ladder is defined by coupling two chains of spins one-half through

the Hamiltonian

$$H_{\text{FL}} = \sum_i J S_{i,a} \cdot S_{i,b} + J'(1+\delta)(S_{i,a} \cdot S_{i+1,b} + S_{i,b} \cdot S_{i+1,a})$$
$$+ J'(1-\delta)(S_{i,a} \cdot S_{i+1,a} + S_{i,b} \cdot S_{i+1,b}). \tag{1}$$

The first index in $S_{i,a}$ denotes the rung number, and the second corresponds to the upper or lower leg. The different couplings are also visualized in Fig. 1. For all calculations in this paper the parameters are set to $J = 0.5$ and $J' = 1$. This model is one member of a family of non-integrable models where geometrical frustration leads to local total spin conservation $\mathbf{S}_{i,a} + \mathbf{S}_{i,b}$ on each rung so that singlet ($S = 0$) and triplet ($S = 1$) configurations are good quantum numbers.

For our discussion, we introduce the following notations. Each rung $r_i$ is either in a triplet state denoted by $|+\rangle$, $|0\rangle$ or $|-\rangle$ corresponding to $S^z_{r_i} = S^z_{i,a} + S^z_{i,b} = +1, 0, -1$ respectively, or in the singlet state $|S\rangle$, with $S^z_{r_i} = 0$. The conserved sectors are labeled by sequences of singlets and triplets on rungs. When a singlet appears on a given rung, the effective coupling with neighboring rungs vanishes. So, in summary, geometrical frustration leads to fragmentation into conserved sectors of singlets and triplets. Then, the presence of singlets further fragments each sector into chains of triplets that are mutually decoupled but which exhibit nontrivial dynamics, given by the dynamics of the spin one Heisenberg chain with coupling strength $J'$. In other words, the Hamiltonian in the rung singlet/triplet basis is block diagonal (cf. Fig 2). These independent subspaces are uniquely determined by the arrangement of triplet and singlet sectors and the magnetization in each connected triplet sector. To label them, each triplet run of length $n$ and magnetization $m$ is denoted by $T^n_m$, and a singlet run of length $n$ by $S^n$. For example, $T^2_0 S$ describes two connected triplets with magnetization $S^z = 0$, followed by one singlet. The structure of the Hilbert space fragments for a ladder of three rungs with $S^z_{tot} = 0$ together with the labeling of the fragments is shown in Fig 2, where the top panel shows the block structure of the Hamiltonian, and the graph in the bottom is the Hamiltonian connectivity graph (corresponding to the adjacency matrix of the Hamiltonian) of basis states of the Hilbert space (in the rung singlet/triplet basis). This representation makes it obvious that there is a large number of disconnected subgraphs corresponding to the Hilbert space fragments. The largest fragment is the sector with triplets on all rungs.

The Hilbert space dimension is $4^L$ where $L$ is the number of rungs while the all-triplet sector, which has dimension $3^L$, corresponds to the non-integrable spin one Heisenberg chain [43]. This sector is expected to exhibit conventional thermalizing behavior. But because this sector is exponentially small compared to the total Hilbert space and because the total Hilbert space is dominated by short chains of triplets separated by singlets, the model overall exhibits non-thermalizing dynamics. In fact, there is an emergent length scale – the typical length of triplet fragments – that leads to localized dynamics as we explain in detail below.

The additional parameter $\delta$ in Eq. (1) tunes the model away from the fully frustrated limit, breaking the local conservation laws and thus leading to a Hilbert space graph with no disconnected components (after reducing to the total $(S, S_z) = (0, 0)$ sector).

In this paper, we consider the generic dynamics of the fully frustrated ladder Eq. (1) using two different types of initial states. The first is a family of product states, where each bond is prepared in the state

$$T(\alpha) = \sqrt{1-\alpha}\,|S\rangle + \sqrt{\alpha}\,|0\rangle. \tag{2}$$

This allows us to systematically consider the effect of the constrained dynamics by tuning the weight of each fragment of the Hilbert space. For $\alpha \to 0$, the singlet states dominate, generating a strongly constrained dynamics while, for $\alpha \to 1$, the dynamics is described by the all-triplet sector and is expected to be fully chaotic. The state for $\alpha = \frac{1}{2}$ is special as it is

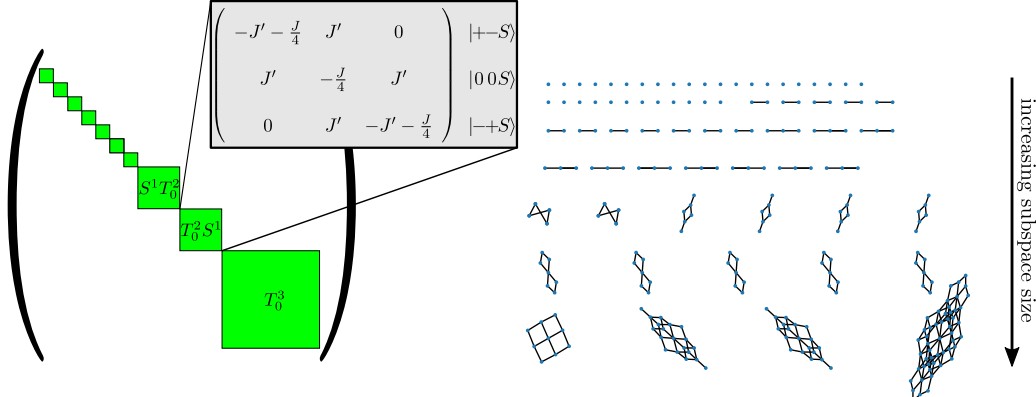

Figure 2: Top: The Hamiltonian structure for three rungs and full frustration ($\delta = 0$) in the $S^z_{tot} = 0$ sector, together with the labels of the largest blocks. The grey inset shows the $3 \times 3$ matrix of the block $T_0^2 S$ in detail.

Bottom: Connectivity graph of the Hamiltonian for $L = 5$ rungs. Each dot corresponds to a basis state in the rung singlet/triplet basis and the connections show the terms of the Hamiltonian. Disconnected subgraphs correspond to the Hilbert space fragments. The largest subgraph is the all triplet sector.

a simple product state of spins $T(\frac{1}{2}) = |\uparrow\downarrow\rangle$ and not merely a product state over rungs. The second class of initial states we consider is the set of Haar-random states.

In the next section we explore the dynamics of the entanglement entropy showing periodic recurrences that can be attributed to short distance physics from short connected triplet segments across the subsystem boundary. We show that the maximum entropy reached during this time evolution can be understood in terms of eigenstate thermalization within the conserved sectors, which leads to an effective independent randomization of the wavefunction in each sector, preserving the weight of the sector. In Section 4 we turn our attention to the quench dynamics viewed through the lens of an out-of-time-ordered correlation function (OTOC). Finally, we lift the frustration (Section 5) with parameter $\delta$ and show how this modifies the entropy growth at short times. In the following sections, the number of rungs is denoted by $L$. It should be emphasized, that two spins are located on each rung, i.e $L = 12$ rungs corresponds to a system with 24 spins.

# 3  Entanglement Entropy and Eigenstate Thermalization

We begin by considering the entanglement dynamics following a quench from the product state

$$|\alpha\rangle = \bigotimes_{i=1}^{L} |T(\alpha)\rangle_i. \tag{3}$$

Fig. 3 shows the entanglement entropy

$$S = -\operatorname{tr}\rho_L \ln \rho_L \tag{4}$$

for an equal bipartition of the ladder in the left half $i = 1 \ldots L/2$ and the right half $i = L/2+1 \ldots L$ for $\alpha = 0.5$.

Using time evolving block decimation [57], we simulate the exact dynamics of the wave function and show the resulting entanglement entropy for system sizes up to $L = 64$ rungs up to bond dimension $\chi = 512$. Relatively low bond dimensions are sufficient here due to the low entanglement generated in the system as a result of the local fragmentation physics.

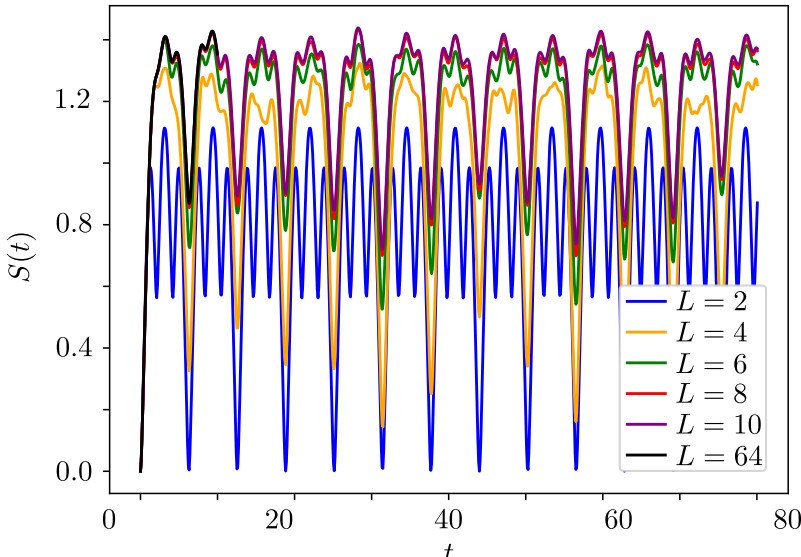

Figure 3: Time evolution of the entanglement entropy for $L = 2 \ldots 64$ rungs after a quench from an initial product state of the form given in Eq. (2) with $\alpha = 0.5$. The results are obtained by exact diagonalization for $L \leq 10$ (up to 20 spins) and TEBD for $L = 64$ (only up to t=10). The revivals in the time evolution are clearly visible even for long times and the maximum entropy saturates for $L \gtrsim 6$. The maximal entropy reached in the course of time is converged with system size and much smaller than the Page entropy expected in ergodic systems ($S_{\text{Page}}(L) = L \ln 2 - \frac{1}{2}$, which is about 43.86 for $L = 64$).

One can observe periodic dips in the entropy with a time period of $2\pi$, independent of the system size. This can easily be understood by means of the energy spacings in small triplet sectors. As discussed in the introduction, the Hilbert space is dominated by singlet-triplet sectors composed of short triplet runs and this weighting towards short triplet fragments is enhanced by taking larger values of the initial state singlet probability $1 - \alpha$ (here 1/2). The energy spacing in the smallest two- and three triplet sectors is commensurate and, with our choice $J' = 1$, the energies are also integer-valued in the two and three triplet sectors, as well as some four triplet sectors (see also the appendix A).

Therefore, within sectors with short triplet runs crossing the subsystem boundary, there is an exact coherent revival of the wavefunction with a period of $2\pi$, leading to a strong reduction of the entanglement entropy. In other sectors with significantly longer triplet runs across the boundary the wave function is scrambled and no such revivals happen due to incommensurate energy differences, but these sectors have a small weight determined by the initial state. Therefore the entanglement production is severely limited by the Hilbert space fragmentation. There are further eigenstates with integer energies in larger sectors. The most significant such states are in the four-rung sector, where these states have an overlap of roughly 50% with the considered initial states.

Turning now to the maximum value of $S(t)$ that exhibits a leading area law scaling of the maximum entropy (constant for large enough $L$ in Fig. 3). This is in clear contrast to the case of fully connected Hilbert spaces, where one expects the entropy to reach the Page value [58] (volume law $\propto L$) at long times. In order to understand this phenomenon, we conjecture that each sector of the Hilbert space thermalizes separately.

Using this assumption, we can estimate the expected maximal entanglement entropy. To do this we consider wavefunctions that are randomized within each conserved subsector but where the overall weight of each such sector is given by the weight in the initial state, since

this cannot change due to the Hilbert space fragmentation. Taking the average entropy over 1000 such random states for different values of $\alpha$ (100 for $L = 8$ and $L = 10$), we compare the results from random properly weighted wavefunctions to the numerically obtained maximum entropy in the course of the time evolution for the initial states, Eq. (2). The results are shown in Fig. 4. Evidently larger Hilbert space fragments thermalize in a conventional manner. This result stands in contrast to, for example, the asymptotic states found for large nonintegrable subsectors of Ref. [39] where even the mid-spectrum states are far from random.

In the case of the initial condition Eq. (2), the probability for a triplet run of $n$ consecutive triplets is equal to $\alpha^n$. It is therefore instructive to introduce the initial condition dependent localization length:

$$l_\alpha = -\frac{1}{\ln(\alpha)}. \tag{5}$$

The maximum entropy scales linearly with the localization length for small $l_\alpha \gtrsim 1$. One can now understand that the violation of the volume law for distinct initial states comes from the fact that the system is only entangled over a distance of order $l_\alpha$. For values $l_\alpha \lesssim \frac{L}{2}$, the effective system size is below the actual length of the ladder and the system obeys an area law, as is also visible in the inset of Fig. 4(a). When, instead, $l_\alpha > \frac{L}{2}$, the entanglement spreads over the entire system and one can detect a crossover to a volume law.

It is important to note that the reduced density matrix for the initial state, Eq. (2), has off-diagonal blocks corresponding to coherences between different conserved sectors that originate from the choice of initial state. These off-diagonal blocks would decohere in a generic thermalizing system but are kept from doing so here because of the local conserved quantities. In fact, these coherences have a sizeable effect on the bipartite entanglement entropy. Indeed, as is shown in the appendix D, while neglecting the off-diagonal blocks would also give an area law for the entropy, the derived bound would overestimate the obtained results by a factor of roughly 1.3 for $\alpha = 0.5$.

The average of the random states deviates from the maximum entropies for small values of $\alpha$ and small system sizes. In this regime the contributions of small sectors dominate. For such small blocks the approximation by a random state is no longer well justified, explaining the deviations. In this regime, the contributions of larger sectors are negligible and the entropy is well approximated by the results for the $L = 2$ chain. This case is also exactly solvable. The first order term for small $\alpha$ gives a logarithmic contribution

$$S_{\max}(\alpha) = \left( 2.365 - \frac{16}{3} \ln \alpha \right) \alpha^2 + \mathcal{O}(\alpha). \tag{6}$$

A comparison to our numerical calculation shows excellent agreement (see appendix C and Fig. 12). Notwithstanding, for larger $\alpha$ and large system sizes the randomized sector wavefunctions give an accurate bound for the maximally reached entanglement entropy.

This picture gives also an accurate bound for the depth of the dips at multiple times of $2\pi$: at these times, the smallest sectors are fully disentangled. As a first estimate, one can now assume the sectors up to length four to be again in the disentangled initial configuration. For sectors up to length three this statement is exact, for four rungs this is a rough, but good estimate. The other sectors are still assumed to be fully scrambled. One can now take again the average over random configurations, but now with the constraint that the triplet subsectors up to length four are fixed to their initial configuration. As can be seen in Fig. 4(b), this approximation recovers the minima of the entropies quite well.

Although the numerics above was restricted to the von Neumann entropy, the observed phenomena also apply to higher-order Renyi entropies and could be thus observed in experiments using randomized measurements [59].

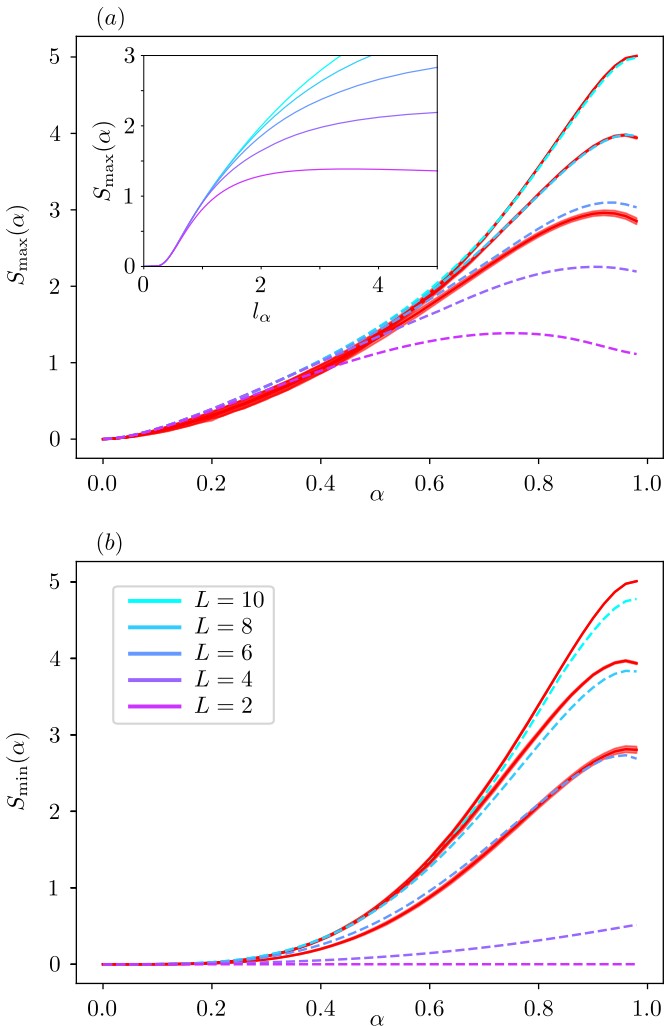

Figure 4: (a): The maximum entropy for different $\alpha$ and the prediction of the random sector approach (red solid lines for $L = 6, 8, 10$). The maxima for the different initial conditions were obtained by time evolution up to t=100.
Inset: The same data, but plotted against the localization length $l_\alpha$.
(b): The minimum entropy after the first maximum at $t = \pi$. The dashed lines show predictions from the random sector approach. The red shaded areas indicate one standard deviation in the distribution of the random configurations.

# 4 OTOC

The entanglement entropy lacks information about the spatio-temporal process of quantum information scrambling. In order to resolve this, we now turn our attention to the out-of-time-ordered correlator (OTOC), which quantifies the spatial spreading of Heisenberg operators. Concretely, we consider the normalized Frobenius norm of the commutator of a spreading operator $S_{a,2}^z(t)$ on the second rung of the $a$ leg of the ladder, with a static "probe" operator $S_{a,i}^z$

$$C(i, t) = \frac{1}{\dim(\mathcal{H})} \|[S_2^z(0), S_i^z(t)]\|_F^2 = -\frac{\text{tr}\left([S_2^z(0), S_i^z(t)]^2\right)}{\dim(\mathcal{H})}. \tag{7}$$

The precise choice of operator, (here $S^z$), affects the details of the OTOC but not the qualitative behavior coming from Hilbert space fragmentation.

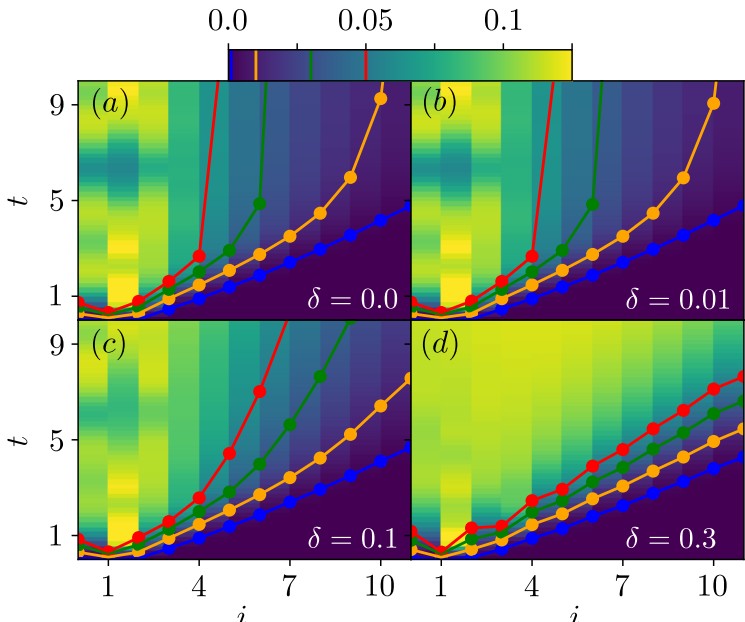

Figure 5: Spacetime evolution of the OTOC $C(i,t)$ Eq. (10) for $\delta = 0.0$ (a), $\delta = 0.01$ (b), $\delta = 0.1$ (c) and $\delta = 0.3$ (d). The curves correspond to contour lines for different thresholds, indicated in the color bar. A sub-ballistic growth, caused by Hilbert space shattering is visible for small $\delta$.

A few general observations about the behavior of OTOCs are useful at this point. Spatially separated local operators commute. As operators evolve in time their support in real space increases. In typical thermalizing systems, this happens in such a way that the commutator in Eq. (7) is significant within the so-called linearly spreading 'lightcone' and exponentially small outside.

The OTOC corresponds to an infinite temperature expectation value and can be calculated using arguments of dynamical typicality [60–62], by replacing the trace in Eq. (7) by an expectation value in a random wavefunction, sampled from the Haar measure. Averaging over a small number of such wavefunctions is sufficient for large Hilbert space dimensions, since the variance over random states is exponentially small in system size. Therefore, we calculate

$$C(i,t) = -\langle \psi | [S^z_{2,a}(0), S^z_{i,a}(t)]^2 | \psi \rangle . \tag{8}$$

Here the operator $S^z_{i,a}$ is evaluated on the $a$ leg of the $i$-th rung. This operator is special in our system, since it swaps the $m = 0$ triplet and the singlet state on the rung, but we expect this to have little effect on the long distance, long time behavior of the OTOC, since this is mainly determined by the existence of local symmetries. A further example is shown in the appendix F. In addition to calculating the standard OTOC (Eq. (7)), it is interesting to consider expectation values as in Eq. (8) over the $\alpha$ product states of Eq. (2), which provides a handle on the dominant length scale of triplet rungs in the system, as explained in the previous section. We will call this quantity $\alpha$-OTOC $C_\alpha(i,t)$ in the following:

$$C_\alpha(i,t) = \langle \alpha | [S^z_2(0), S^z_i(t)]^2 | \alpha \rangle . \tag{9}$$

In our system, by expanding the commutator, the OTOC simplifies to

$$C(i,t) = \frac{1}{8} - 2\,\mathrm{Re}\,\langle \psi | S^z_2(0) S^z_i(t) S^z_2(0) S^z_i(t) | \psi \rangle . \tag{10}$$

To calculate this expression numerically, we compute the overlap of the states $S^z_{2,a}(0)S^z_{i,a}(t)|\psi\rangle$ and $S^z_{i,a}(t)S^z_{2,a}(0)|\psi\rangle$. These states can be determined using Krylov space time evolution techniques [61, 63, 64], and here we show results for up to 12 rungs (24 spins).

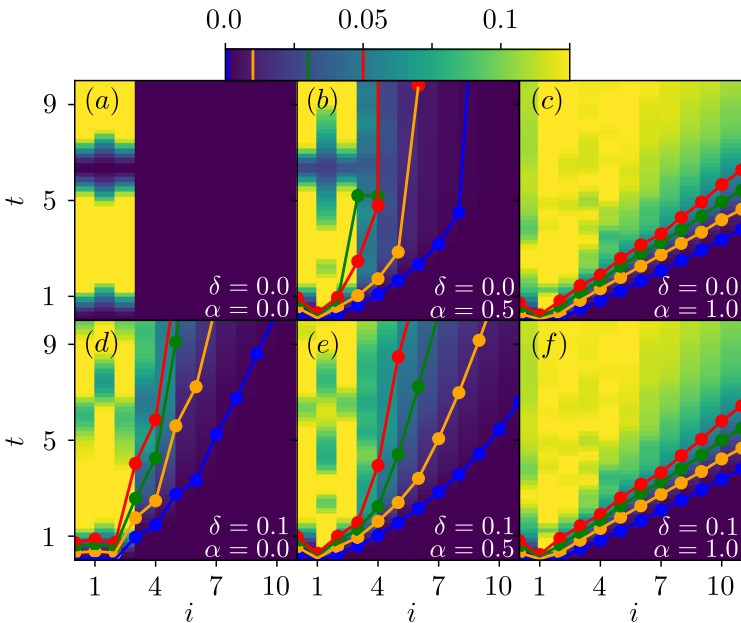

Figure 6: Spacetime evolution of the $\alpha$-OTOC $C_\alpha(i,t)$ Eq. (9) for $\alpha = 0.0$ (a,d), $\alpha = 0.5$ (b,e) and $\alpha = 1.0$ (c,f), for $\delta = 0.0$ (a-c), and $\delta = 0.1$ (d-f), $L = 12$ rungs. The colored lines correspond to contours for thresholds indicated in the colorbar.

The results for the full OTOC, Eq. (7), are shown in Fig 5. Again one can observe fingerprints of the small Hilbert space fragments with integer energy eigenvalues, leading to oscillations with periods $T \sim 2\pi$, which survive also small perturbations from full frustration. These small Hilbert space fragments correspond to small runs of triplets in real space and hence the oscillations are only coherent at short distances, confirming our picture.

More remarkable is the apparent sub-linear light cone front of the OTOC. The reason for the suppression of the operator spreading is more visible for the $\alpha$-OTOC $C_\alpha(i,t)$ Eq. (9) (see Fig 6). In the pure singlet case ($\alpha = 0$), the dynamics is fully blocked beyond $i = 2$. On the other side, the triplet state $\alpha = 1$ shows ballistic light cone spreading which is expected for thermal systems.

Both observations can be explained by the fragmentation into local chunks of the Hilbert space. Since the dynamics across a singlet rung is blocked, the OTOC $C(i,t)$ has only contributions from terms with triplet chains connecting rungs 2 and $i-1$. As a first approximation, one supposes that each of these contributions scrambles in the given sector. If this assumption is valid, the OTOC spreading should be equivalent to the behavior of thermal systems, however with an accessible dimension of the Hilbert space which depends on the distance between the spreading and the probing operator. This accessible fraction of the Hilbert space is $\propto \alpha^{i-2}$ (see also Appendix E). In the case of Eq. (9), this yields the scaling behaviour

$$C_\alpha(i > 1, t) = \alpha^{i-2}C_{\alpha=1}(i,t). \tag{11}$$

The $\alpha$-OTOCs for different values of $\alpha$ and the predictions Eq. (11) are shown in Fig 7. For large distances the curves show perfect agreement.

Apart from this distance-dependent rescaling, the long-distance behavior of the OTOCS can be fully explained by known results for short times using perturbation theory [64] and for random unitary circuits at intermediate times [65, 66].

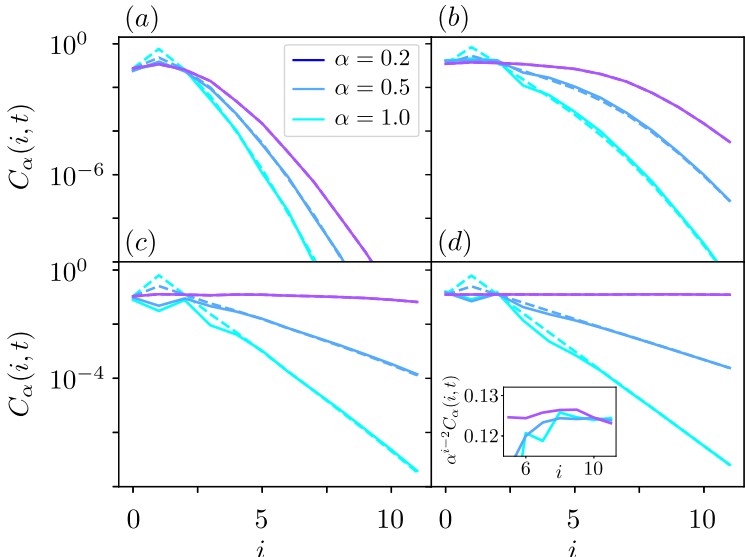

Figure 7: Fixed time $\alpha$-OTOCs $C_\alpha(i, t)$ ($L = 12$), numerical results (solid) and their predicted scaling Eq. (11) from the Hilbert space fragments (dashed lines) at time $t = 1.0$ (a), $t = 3.0$ (b), $t = 7.0$ (c) and $t = 30.0$ (d). Inset: The same Otoc, now rescaled by $\alpha^{i-2}$.

Using a perturbative expansion for the time evolution operators at short times, one can show the $\alpha$-OTOC $C_\alpha(i, t)$ to grow by a distance dependent power law as [64]

$$C_\alpha(i > 1, t) = \alpha^{i-2} \frac{t^{2(i-1)}}{(i-1)!} \mathcal{O}(1). \tag{12}$$

As can be seen in Fig. 8(a), this is in agreement with the numerical obtained results.

For intermediate times, one can compare the results with predictions of random unitary circuits [65]

$$F(i, t) \simeq 0.125 \alpha^{i-2} \, \mathrm{erfc}\left( \frac{(i-1) - v_b t}{\sqrt{2t(v_l^2 - v_b^2)}} \right). \tag{13}$$

Here $v_l$ and $v_b$ denote the so-called lightcone and butterfly velocity. Fig. 8(b) shows the convergence towards the saturation value $C_\alpha(i, \infty)/\alpha^{i-2} = \frac{1}{8}$ and the comparison to the random unitary circuit predictions, Eq. (13), with $v_b = 1.5$ and $v_l = 1.9$. The results show reasonable agreement for intermediate times. At longer times one can see deviations. This is expected to be caused by hydrodynamic tails [66]. Indeed, at long times, the difference between the infinite time limit and the $\alpha$-OTOC is expected to decay as $\propto 1/\sqrt{v_b t - (i-1)}$, as is indicated by a black dashed line in the plot.

We have seen generic long distance tails and intermediate scale exponential decay coming from fragmentation. A further fingerprint of locality of fragments are the oscillations at short distances. In this regime the contributions of small sectors dominate and the approximation of fully scrambled sectors is no longer valid. The oscillations originate from revivals in the smallest subspaces. To verify this, we consider now the full OTOC, Eq. (7) (Frobenius norm), and sum up the contributions of sectors with connected triplet rungs up to length $l$, denoted by $C^l(i, t)$. This is shown in Fig. 9. For instance, the two triplet term for the case $C^{l=2}(1, t)$ contains all contributions where only one rung neighboring $i = 1$ is in a triplet state. The rung $i = 1$ is undetermined, i.e. it can be either in a triplet or singlet state, since the evaluation

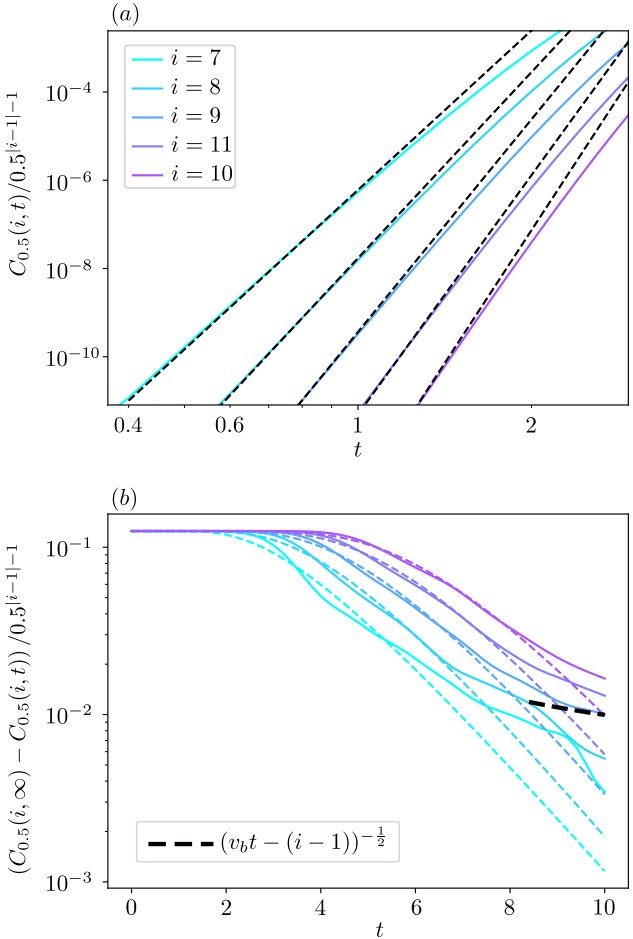

Figure 8: (a) Rescaled $\alpha$-OTOC growth ($L = 12$) for fixed distance $i$ at early times, fitted by a power-law scaling Eq. (12) (dashed lines). (b) Time evolution of the rescaled $\alpha$-OTOC at intermediate times. The dashed lines indicate predictions for random unitary circuits Eq. (13). At long times one expects hydrodynamic tails due to charge conservation [66], indicated by a black dashed line.

of the operator $S_i^z$ can flip triplets and singlets. The contributing states are in this case of the form $|TUS\ldots\rangle$ or $|SUTS\ldots\rangle$, where $U$ denotes the undetermined rung at $i = 1$.

We clearly see that there is a rapid convergence from short triplet chains connecting the two operators towards the full result (over all contributions, black line) at short distances, while for longer distances longer triplet chains are needed.

Longer sectors are approximated as thermal, described by the long time value $\frac{1}{8}$. This is a rough approximation, which is not correct at short times, but sufficient to explain the role of the small sectors and their impact on oscillations in the OTOC. As can be seen in the upper panel of Fig. 9, already the exact summation up to triplet sectors of length 3 recovers all short distance oscillations. The isolated contributions of different sector lengths are shown in the lower panel of Fig. 9. Sectors with at least four triplets show long time saturation, such that the thermal approximation for such sectors is justified retrospectively. As is clear from the construction of the different sectors, triplet sectors of length $i$ can only contribute to OTOCs with distance of at most $i-1$ between the two applied operators. In other cases, the dynamics is blocked by a singlet between the two applied operators. Thus the two and three-triplet sectors do not contribute at larger distances, which explains the absence of their signatures in this regime.

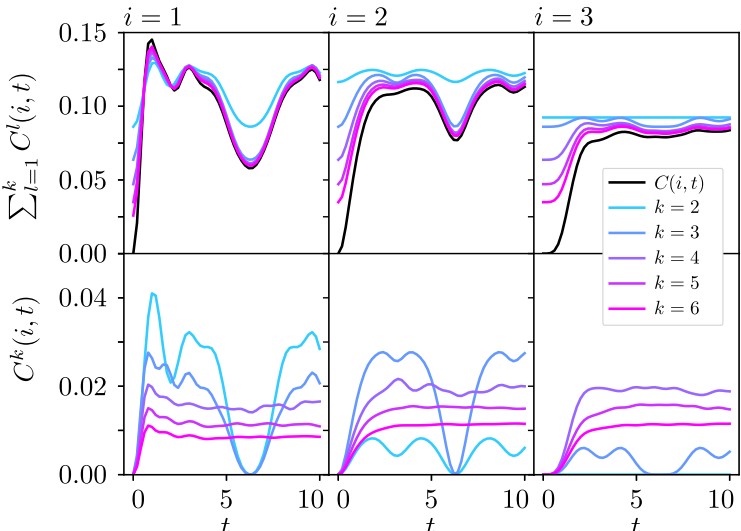

Figure 9: Contribution of the different $k$-triplet sectors for $C(i,t)$. The upper row shows the OTOC approximations $\sum_{l=1}^{k} C^l(i,t)$ by respecting triplet sectors up to length $k$ together with the numerical obtained OTOCs (black) for $L = 12$ rungs. The bottom row shows the isolated contributions $C^k(i,t)$ of the different sector lengths.

# 5 Lifted Frustration

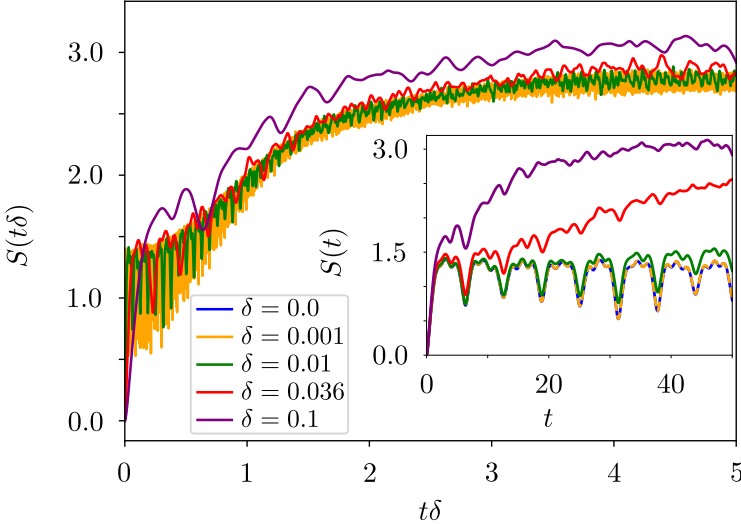

Figure 10: Rescaled entropy growth for $\delta = 0.001$ (orange), $\delta = 0.01$ (green), $\delta = 0.036$ (red), $\delta = 0.1$ (purple) and L=6 rungs. The time is rescaled by the detuning factor $\delta$.

Inset: The entropy growth, now presented with the original time scale and in comparison to the fully frustrated case $\delta = 0$ (blue). One can observe perfect phase locking of the revivals.

By tuning away from full frustration, the strong constraints to the dynamics due to the extensive number of conserved quantities are lifted. This can be also observed in the entropy growth, where now the expected linear growth proportional to the detuning parameter $\delta$ is visible, as can be seen in Fig. 10. However, one can still observe recurrences in the entropy time evolution, until a saturation value is reached. Thus, signatures of the underlying strongly

fragmented Hilbert space affect the dynamics also away from this special point.

These revivals can also be observed in the OTOCs, at least for small $\delta \lesssim 0.1$, see e.g in Fig. 5. For larger detuning $\delta \gtrsim 0.1$, all signatures of the fully frustrated limit are washed away and one can finally detect the crossover to ballistic spreading of the light cones.

The main result of Fig. 10 is that there is a collapse of the computed entropy growth with respect to rescaled time $t\delta$. This implies that there should be saturation to fully thermalized behavior for arbitrarily small $\delta$ in the long time, large system size limit.

Even if the nonergodic behaviour is unstable to detuning of all coupling parameters, it should be noticed that the system is stable with respect to single local perturbations, such that only a few of the local conserved quantities are affected.

Another interesting direction is to consider other degeneracy breaking terms. For example, the existence of local magnetic fields would also destroy the exact local conservation laws. However, in this case one has residual so-called dynamical symmetries [67]. These lead to persistent oscillations in the time evolution of operators having finite overlap with this symmetries. In contrast to the fully frustrated model discussed in this work, the frequency of those oscillations is no longer determined by the coupling parameters, but instead by the magnetic field strength.

# 6 Summary and Conclusions

To summarize, we have obtained a full qualitative and quantitative understanding of entanglement dynamics and operator spreading in the fully frustrated ladder. This system is known to exhibit a fragmentation of the Hilbert space into exponentially many disconnected sectors. The central observation of this work is that large enough fragments thermalize in the sense of obeying the eigenstate thermalization hypothesis but the Hilbert space is exponentially dominated by short chain fragments that are far from thermal. The net result is the appearance of periodic oscillations in the entropy and in the OTOC at short distances coming from the simple discrete spectra of very short chain fragments. Fragmentation also leads to an emergent length scale and dynamical localization that we clearly observe at intermediate length and time scales in the operator spreading.

As a corollary, these results apply equally to kinetically constrained or fractonic systems where shattering into exponentially many disconnected subspaces is tied to spatial locality. In particular one expects to observe recurrences in the dynamics in such settings though these are particularly pronounced in our model as the short chain fragments have regular level spacings which also makes it attractive for experimental realizations. One also expects to observe localization of information spreading when local fragments are exponentially populous in the space of states.

Finally, we also investigated the fate of the system with lifted constraints, finding that signatures of the fragmented Hilbert space influence the dynamics of the system at short times. For longer times, ETH seems to be restored for any finite detuning $\delta$.

# Acknowledgments

We thank Masudul Haque, Johannes Richter and Arnab Sen for related collaborations. We also thank Berislav Buča for valuable discussions. We acknowledge financial support from the Deutsche Forschungsgemeinschaft through SFB 1143 (Project-id 247310070).

Table 1: Energy eigenvalues in the smallest triplet sectors $T_i^\ell$ of triplet runs up to length $\ell = 3$ in the fully frustrated ladder ($\delta = 0$) with $J' = 1$. All eigenvalues are integers, leading to coherent oscillations in the OTOC and entanglement dynamics.

| sec. | $E_1$ | $E_2$ | $E_3$ | $E_4$ | $E_5$ | $E_6$ | $E_7$ |
|------|------|------|------|------|------|------|------|
| $T_0^1$ | 0 | | | | | | |
| $T_1^1$ | 0 | | | | | | |
| $T_0^2$ | $-2$ | $-1$ | 1 | | | | |
| $T_1^2$ | $-1$ | 1 | | | | | |
| $T_2^2$ | 1 | | | | | | |
| $T_0^3$ | $-3$ | $-2$ | $-1$ | $-1$ | 0 | 1 | 2 |
| $T_1^3$ | $-3$ | $-1$ | $-1$ | 0 | 1 | 2 | |
| $T_2^3$ | $-1$ | 1 | 2 | | | | |
| $T_3^3$ | 2 | | | | | | |

## A  Energy levels of the smallest sectors

Table 1 shows the energy eigenvalues in the fully frustrated case $\delta = 0$ for the smallest triplet sectors, with $J' = 1$, as used in the main text. The sectors are labeled by the convention introduced in 2. The eigenvalues for positive and negative magnetization sectors are identical. In the sectors shown in the table, all eigenvalues are integers, leading to strong coherent oscillatory contributions to the entanglement and OTOC dynamics at short distances. We note that integer eigenvalues are present sporadically in larger sectors, but there are no further all-integer eigenvalue sectors apart from those listed here.

## B  Convergence of the TEBD computations

The results in Fig. 3 were obtained by a first order Trotter decomposition, with time step $dt = 0.01$ and maximal bond dimension $\chi = 512$. As can be seen in Fig. 11, these parameters are sufficient to get a good convergence of the results.

## C  Estimations for the maximum entropy for small $\alpha$

In the case of small $\alpha$, Fig. 4 reveals deviations from the linear growth of the maximal entropy with the localization length. Due to the short effective length scale, one can approximate the entropy time evolution in this regime by the results for $L = 2$ rungs, since contributions of larger sectors become negligible. This case is analytically solvable. To simplify the notations, we neglect additional phase factors caused by the coupling strength $J$, since these cancel out in the computation of the entropy, $J' = 1$ as before.

The only nontrivial sector in this case is $T_0^2$. For the initial state $|00\rangle$, the time evolution is given by

$$|\psi_0(t)\rangle = f_1(t)|00\rangle + f_2(t)|+-\rangle + f_2(t)|-+\rangle ,\tag{14}$$

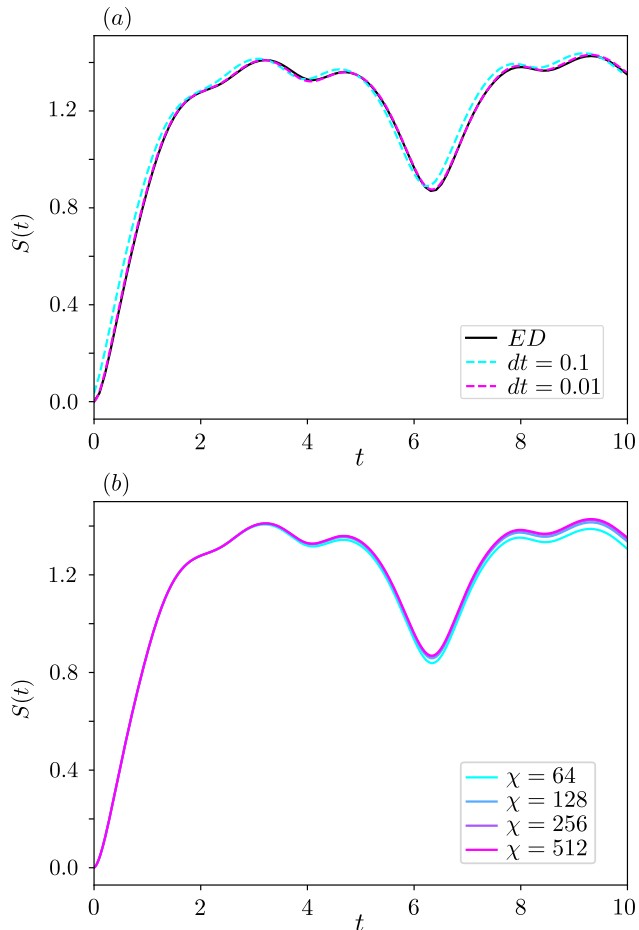

Figure 11: (a) Comparison of the entropy time evolution for L=10 rungs, obtained by exact diagonalization (black) and TEBD using a first-order Trotter decomposition, with time steps $dt = 0.01$ and $dt = 0.1$
(b) Results for $L = 64$ rungs and $dt = 0.01$ with different maximal bond dimensions $\chi$.

with $f_1(t)$ and $f_2(t)$ given by

$$
\begin{aligned}
f_1(t) &= \frac{1}{3}(-e^{2it} + 2e^{-it}), \\
f_2(t) &= \frac{1}{3}(e^{2it} - e^{-it}).
\end{aligned}
\tag{15}
$$

The the state $|\alpha(t)\rangle$ has the form

$$
\alpha(t) = \sum_{i,j\in\{S,0,+,-\}} M_{i,j}(\alpha,t)\,|i\rangle\,|j\rangle .
\tag{16}
$$

Here $|i\rangle$ ($|j\rangle$) denotes the state of the left (right) rung. The matrix $M$ has the following coefficients:

$$
M(\alpha,t) = \begin{pmatrix} 1-\alpha & \sqrt{\alpha(1-\alpha)} & 0 & 0 \\ \sqrt{\alpha(1-\alpha)} & \alpha f_1(t) & 0 & 0 \\ 0 & 0 & 0 & \alpha f_2(t) \\ 0 & 0 & \alpha f_2(t) & 0 \end{pmatrix}.
\tag{17}
$$

The reduced density matrix is then given by

$$\rho_{\text{red}}(\alpha, t) = M(\alpha, t)M^\dagger(\alpha, t). \tag{18}$$

In the case of $L = 2$, the maxima of the entropy are exactly at odd multiples of $t = \pi$. In this case the eigenvalues of $\rho_{\text{red}}(\alpha, t)$ are given by

$$
\begin{aligned}
\lambda_1 &= \frac{4\alpha^2}{9}, \\
\lambda_2 &= \lambda_1, \\
\lambda_3 &= \frac{1}{18}\left(9 - 8\alpha^2 - \sqrt{(3-4\alpha)^2(9+24\alpha-32\alpha^2)}\right), \\
\lambda_4 &= \frac{1}{18}\left(9 - 8\alpha^2 + \sqrt{(3-4\alpha)^2(9+24\alpha-32\alpha^2)}\right).
\end{aligned}
\tag{19}
$$

The entropy can be obtained by

$$S_\alpha = \sum_{i=1}^{4} -\lambda_i \ln \lambda_i. \tag{20}$$

In the case of small $\alpha$ this reduces to

$$
\begin{aligned}
S_{\text{max}}(\alpha) &= \left(\frac{8}{3} - \frac{20}{9}\ln\frac{16}{9} + \frac{4}{9}\ln 9 - \frac{16}{3}\ln\alpha\right)\alpha^2 + \mathcal{O}(\alpha^3) \\
&\simeq \left(2.365 - \frac{16}{3}\ln\alpha\right)\alpha^2 + \mathcal{O}(\alpha^3).
\end{aligned}
\tag{21}
$$

The comparison between the maximum entropy and Eq. (21) is shown in Fig. 12. The curves show perfect agreement for small $\alpha$.

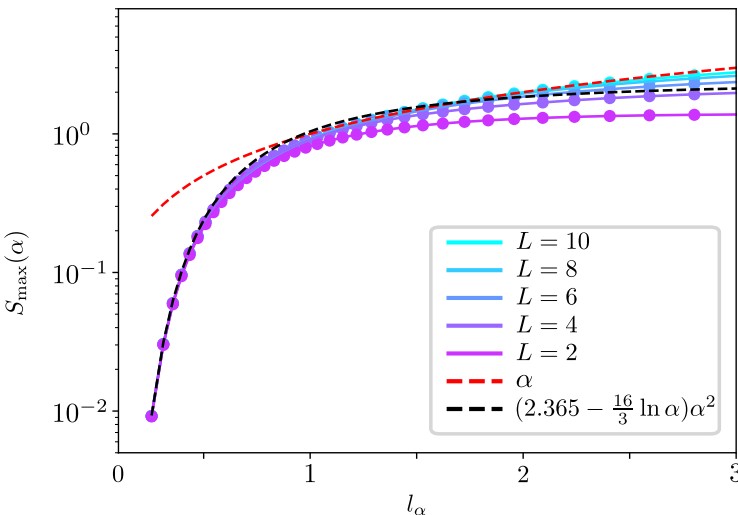

Figure 12: The maximum entropy for small $\alpha$, and the comparison to Eq. (21).

# D  Block diagonal approximation for the entropy

The following section provides a naive estimate for the maximum bipartite entanglement entropy of the state Eq. (2) with $\alpha = 0.5$, by ignoring off-diagonal elements in the reduced density

matrix between disconnected sectors. A generalization to other states is straightforward. The purpose of these computations is to emphasize the role of coherences between different disconnected blocks.

For this initial state the probability of a single rung to be in a singlet or in a triplet is equal to $\frac{1}{2}$. Consider now the reduced density matrix of the left subsystem $\rho_l = \mathrm{tr}_r |\psi\rangle \langle\psi|$, after tracing out all degrees of freedom right to the center bond. The state has with probability $P_0 = \frac{1}{2}$ one singlet left to the cut, with probability $P_1 = \frac{1}{2 \cdot 2} = \frac{1}{4}$ one triplet followed by one singlet, and so on. The probability for $k$ connected triplets at the bond, followed by one singlet, is then given by:

$$P_k = \frac{1}{2^{k+1}}. \tag{22}$$

The block diagonal approximation assumes now, that the matrix elements between these different blocks are randomly distributed and thus can be neglected for the estimation of the maximum entropy. Because off-diagonal blocks are neglected, this must give an overestimate for the maximum entropy. We show that, for ten rungs, the entropy is 2.41 compared to the numerical value of about 1.4.

Each $k$-triplet sector contains $N_k = 3^k$ states. As a first approximation, assume that the initial configuration is scrambled among all accessible states within one sector. This means that the probability to be in a specific state with $k$ triplet rungs at the cut is

$$P_k^s = \frac{1}{2^{k+1}3^k}. \tag{23}$$

With the assumption of independent sectors, i.e. a mixed state, one obtains the estimate for the entropy $S_0^{L/2}$ for infinitely long systems ($L/2 \to \infty$)

$$\begin{aligned}
S_0^\infty &= \sum_{k=0}^\infty -N_k P_k^s \ln(P_k^s) \\
&= \sum_{k=0}^\infty \frac{1}{2^{k+1}} \ln(2^{k+1}3^k) \simeq 2.49.
\end{aligned} \tag{24}$$

Here, the subscript indicates the number of non-scrambled triplet sectors (0 in this case). Thus the entropy stays bounded and obeys an area law in the thermodynamic limit. In order to compare to the numerics for 10 rungs, one gets for this finite system $S_0^5 \simeq 2.41$. Comparison with the results in Fig. 3 shows, that obtained bound is an overestimate in comparison with the numerical results.

However, one can slightly improve this estimate. Closer inspection of the time evolution in the different sectors shows that the approximation of fully scrambled 1 and 2 triplet sectors is too rough. Let us assume now the opposite, that these sectors are not scrambled at all. Since for $\alpha = 0.5$ small triplet sectors still dominate, this is not a bad approximation.

Denoting $S_i$ the probability for no scrambling in the $i$ smallest triplet sectors, this can be written as

$$S_i^\infty = -\sum_{k=0}^{k=i} \frac{1}{2^{k+1}} \ln\left(\frac{1}{2^{k+1}}\right) + \sum_{k=i+1}^\infty \frac{1}{2^{k+1}} \ln\left(2^{k+1}3^k\right). \tag{25}$$

Thus one obtains the results

$$\begin{aligned}
S_1^\infty &= \simeq 2.21, \\
S_2^\infty &= \simeq 1.94.
\end{aligned} \tag{26}$$

In the case of 10 rungs, the bound reduces to:

$$
\begin{aligned}
S_1^5 &\simeq 2.13\,, \\
S_2^5 &\simeq 1.86.
\end{aligned}
\tag{27}
$$

We have checked that these bounds are borne out by the numerical simulations by taking the reduced density matrix at the time of maximal entropy, starting from the $|\alpha = 0.5\rangle$ states, and setting the off-diagonal blocks to zero. This yields an entropy of $S \simeq 1.9$, in agreement with the above bound $S_2^5$. This is significantly larger than the maximal entropy observed in the dynamics $S^{\mathrm{max}} \simeq 1.4$ for the $\alpha = 0.5$ state.

This difference is therefore caused by the off-diagonal blocks. A closer look at them reveals that the random distribution of the off-diagonal elements is not entirely correct. Instead, the coherent fragmentation of the Hilbert space also induce correlations between these elements which leads to a decrease of the entropy bound. However, it is difficult to get exact analytical expressions for this effect.

## E   Toy Calculation of the OTOC for the Fully Frustrated Ladder

We present a toy calculation that captures the exponential fall-off of the OTOC for the ladder. We focus on the $\alpha$-OTOC, $C_\alpha(i, t)$ (Eq. (9)) where the reference site for the numerical calculations in the main text is site 2. Hilbert space fragmentation allows us to consider each fragment separately. We assume that the light cone velocity is the same for runs of triplets in all sectors, that blocking of information spreading only occurs because of the presence of a fixed singlet rung so that we neglect the constraint of angular momentum conservation. So far, these are fairly weak assumptions. But we further assume that scrambling is immediate and total once the probe site $i$ is inside the light cone. This assumption is made to capture the pure effects of Hilbert space fragmentation and neglects the otherwise interesting details of many-body operator spreading.

We need to know the number of consecutive triplets in the conserved sector counted from reference site, say, to the right of this site. Call this $n_T$. This is different in general from the total number of triplets in the sector that we call $N_T$. After time $t$ the light cone extends over a distance $vt \equiv p$ rungs of the ladder until it is cut off by the appearance of a singlet. Then there are different cases:

1. If $p < i$, the commutator vanishes as so $C_\alpha(i, t) = 0$.

2. If $p > i$ but $n_T < i$ then again $C_\alpha(i, t) = 0$ because information cannot spread beyond the rung position $n_T$ since there is a singlet bounding it by assumption.

3. The nontrivial case is when $n_T \geq p \geq i$ because the ordering $p > n_T > i$ is forbidden by the fact that $p$ cannot spread beyond the row of consecutive triplets.

Then

$$
\begin{aligned}
C(i < t = p/v) &= J \sum_{n_T = i}^{p} \alpha^{n_T} \\
&= J \frac{1}{1 - |\alpha|} |\alpha|^i \left(1 - |\alpha|^{p-i+1}\right),
\end{aligned}
\tag{28}
$$

with $J$ a constant coming from the evaluation of matrix elements. It follows that $C(i, t)$ is trivial for $\alpha = 0$ and thermalizing for $\alpha = 1$ as we should minimally expect. In addition, the OTOC falls off exponentially with distance at long times as $|\alpha|^i$.

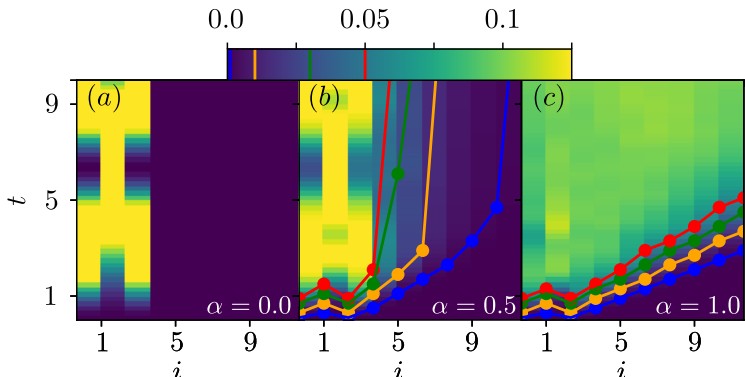

Figure 13: The $\alpha$-OTOC $C_\alpha^{zx}(i,t)$ and L=10. The oscillations at short distances for small $\alpha$ and the long-distance decay are also visible for this OTOC.

## F   OTOCs for the $S_x$ operator

Since the long-range decay of OTOCs is caused by the local symmetries, the phenomenology for the OTOCs described above applies also to other operators than $S_z$. For example, the results of the $\alpha$-OTOC

$$C_\alpha^{zx}(i,t) = \langle\alpha|[S_{2_b}^x(0), S_{i_a}^z(t)]^2|\alpha\rangle , \tag{29}$$

are shown in Fig. 13. The $S_x$ operator is evaluated at the $b$ leg, the $S_z$ operator on the $a$ leg. Since the operators couple to different symmetry sectors, the Otoc at short distances is slightly different. Nevertheless one can see again Rabi oscillations in short-triplet sectors. The long range decay is the same as in Fig. 6.

## G   Accuracy of the shortest contributions

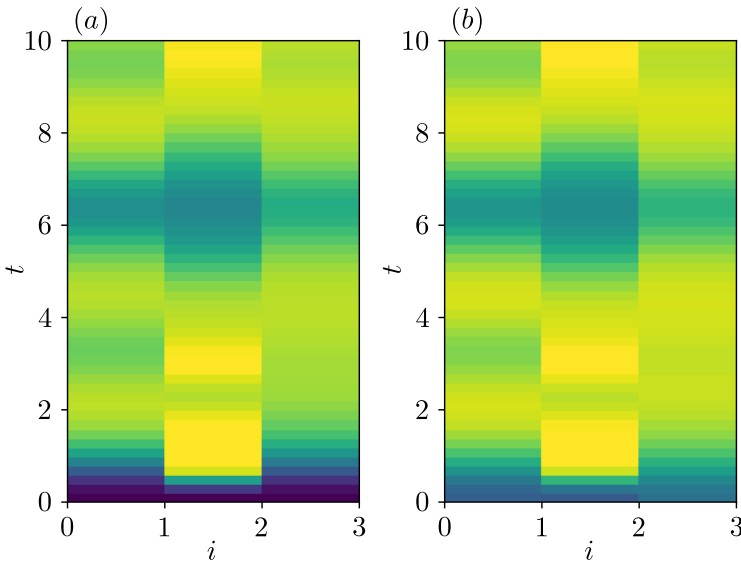

Figure 14: The OTOC $C(i,t)$ at short distances, original data (a) and short term contributions $\sum_{l=1}^{k=4} C^l(i,t)$ (b). Apart from the initial growth, all features are well captured by the contributions of the shortest sectors.

In Fig. 14, an additional comparison between the OTOC $C(i, t)$ at short distances and the contributions of the smallest sectors $\sum_{l=1}^{k=4} C^l(i, t)$ is shown. As discussed in the main text, the initial growth cannot be captured within this simplified picture. Apart from this, all oscillations are captured remarkably well by the shortest sectors.

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
