# Peer review of "Information Dynamics in a Model with Hilbert Space Fragmentation"

_SciPost Physics, doi:SciPost Phys. 11, 074 (2021)_

## Round 2 · Referee Report · Anonymous (Referee 1) · 2021-5-20

Strengths

  1. The paper is timely.
  2. Lucidly and honestly written.
  3. Comprehensive, and apparently correct.
  4. The paper makes modest progress on an existing problem: How does information spread in constrained quantum systems.?

Weaknesses

  1. The results are likely of only limited experimental interest.
  2. While comprehensive, the results of the paper are not greatly surprising.

Report

This paper was a pleasure to read. The results were clearly and honestly presented, and they are reasonably interesting.

A few questions I'd like the authors to answer.

Q1. This hamiltonian simply (up to additive constants) a spin-1 heisenberg model, when acting on a string of triplets (exchange coupling $J'$). Correct? The authors should emphasize this more.

Q2. I think the level spacings within a string of triplets will only depend on $J'$ (not on $J$). In that case, why do the authors say (pg 4, LH column) "recurrences ... should only arise whenever $J,J'$ are chosen so that ... commensurate"? This sentence suggests that $J$ is important to the question of recurrences. But only $J'$ matters, I think.

Q3. The authors examine the von Neumann entropy of time evolved states. Are there any interesting qualitative changes when examining higher Renyi entropies?

Requested changes

  1. Answer Q1,Q2,Q3 above, reflecting answer in the text.
  2. Delete redundant "." at bottom of page 7, right hand column.

  • validity: high
  • significance: good
  • originality: good
  • clarity: top
  • formatting: excellent
  • grammar: excellent

Author:  Dominik Hahn  on 2021-05-21  [id 1446]

(in reply to Report 1 on 2021-05-20)
Category:
answer to question

We would like to thank the referee for a careful reading of our paper and the positive assessment. We appreciate the valuable suggestions. Below, we answer the questions put forward by the referee and outline the main resulting changes to the manuscript.

We agree with the referee that it would be interesting to realize dynamical experiments in the fully frustrated ladder. The situation is perhaps not quite as hopeless as it might seem at first glance. Given the progress in quantum simulation using cold atomic systems and also Rydberg atoms, we are reasonably optimistic that the model we studied might come within reach in the near future.

Q1. This hamiltonian simply (up to additive constants) a spin-1 heisenberg model, when acting on a string of triplets (exchange coupling J′). Correct? The authors should emphasize this more.

The Hamiltonian for connected triplet sectors is indeed given by the Heisenberg chain. We added a sentence in the description of the model to emphasize this statement.

Q2. I think the level spacings within a string of triplets will only depend on J′ (not on J). In that case, why do the authors say (pg 4, LH column) "recurrences ... should only arise whenever J,J′ are chosen so that ... commensurate"? This sentence suggests that J is important to the question of recurrences. But only J′ matters, I think.

The referee is correct. The oscillations in the entropy only depend on J' .We have removed this section.

Q3. The authors examine the von Neumann entropy of time evolved states. Are there any interesting qualitative changes when examining higher Renyi entropies?

The analysis of the von Neumann entropy also applies to higher order Renyi entropies, thus the area law entanglement and the oscillations could in principle be measured in experiments either using replicas or randomized measurements. We have added a paragraph at the end of the third section.

---

## Round 2 · Referee Report · Anonymous (Referee 2) · 2021-5-25

Strengths

  • timely topic
  • clear
  • well written

Weaknesses

  • standard numerics on one example
  • absence of novelty
  • missing comparison with the relevant litterature

Report

The authors of the manuscript study the dynamics of information spreading in a system with Hilbert space fragmentation. They look at the entanglement entropy and the square-commutator dynamics and they link their behaviour to the salient features of the model under consideration. The results are based on a numerical investigation on a frustrated spin 1/2 ladder, where the two couplings (intra-ladder and between nearby spins) are chosen to be commensurate. They also study how their findings change when the specific degeneracy inducing the fragmentation is lifted.

The strength of this manuscript is that it is an interesting and timely topic. The work is located an active trend of the research in quantum information and condensed matter, in particular concerning information spreading in relation to the failure of standard thermalization. Furthermore, the manuscript is clear and well written.

However, the paper does not represent a breakthrough and the results suffer from two major flows.
First of all, the work is based on numerical observations on a single model. The authors make several conjectures and claims of generality, but not all of their statements are supported by accurate numerical tests. The numerics themselves are restricted to small system sizes (up to $L=10$). And, while in one plot they also compare with a block decimation simulation for $L=64$, it is not clear why the same technique is not used for the rest of the results.
Secondly, it is unclear the relevance of this paper compared to similar works in the literature. Despite the introduction is well written in presenting the underlying context, the authors do not make enough comparisons between their results and the other papers on the same topic.

For these reasons, I do not think that this paper meets the criteria of novelty required by SciPost Physics. On the other hand, I would suggest publication in SciPost Physics Core, provided that the issues below are carefully addressed.

Requested changes

-- Main issues--

A1. Comparison with the literature. The authors shall make an effort in explaining what makes their paper interesting to respect to what is already known in other systems displaying Hilbert space fragmentation.

A2. Numerics. Why is the TEBD simulation performed only for the entanglement entropy dynamics for $\alpha=0.5$? Is there a technical reason? The authors shall try to improve the numerics, also using TEBD for larges system size when possible.

A3. The square commutator is studied only for the $\hat S^z$ operator. In the case of chaotic dynamics, the choice of the operator is inessential. However, the model under analysis is a strongly non-ergodic system. In this case, one expects relevant differences, especially if the operator is associated with the conservation law enforcing the fragmentation. It is claimed that the qualitative behaviour of the results is not affected by this choice. The authors shall show at least an example to corroborate this claim, even in the appendix.

A4. The authors say that oscillations (of entanglement and the square-commutator) arise when the couplings $J$ and $J'$ are commensurate. What happens when they are not?

A5. The effect of the lift of degeneracies is an interesting point discussed in the paper. The authors shall comment more on this issue. For instance, is it possible to estimate the saturation point value of the entanglement entropy? How would things change with different degeneracy breaking terms? How would things change with a single impurity? Is fragmentation always unstable?

-- Minor issues, style and details --

B1. Below figure 8, "the sum of the contributions $C^l(i, t)$ of sectors with increasing lengths ..." is introduced. While the meaning of the sentence is quite intuitive, this notation has never been defined. It would be worth to state more clearly the quantities of interests.

B2. If the scope of Fig.7 is to show the scaling $C_\alpha(i>1, t) \propto \alpha^{i-2}$, why the authors did not plot directly that $C_\alpha(i>1, t)/ \alpha^{i-2}$?

B3. The authors say "the maximal entropy scales linearly with the localization length for small $l_\alpha$". The inset of Fig.4a shows a collapse for small $l_\alpha$, which is not exactly linear (probably because of the small system sizes $L$). Is there another plot showing this?

B4. Fig.11: missing x-axis label.

B5. Figs.6-10: missing system size $N$ in the caption.

B6. The colours and lines of Fig.7 are confused. The authors shall try to improve the plot.

B7. The double point at the end of page 7.

  • validity: good
  • significance: ok
  • originality: ok
  • clarity: high
  • formatting: good
  • grammar: good

Author:  Dominik Hahn  on 2021-06-23  [id 1519]

(in reply to Report 2 on 2021-05-25)

We would like to thank the referee for a careful reading of our paper and the valuable suggestions. Below, we answer the questions put forward by the referee and outline the main resulting changes to the manuscript.

We are grateful for the general recommendation for publication of our manuscript after addressing the referee's comment, but respectfully disagree with the assessment that this paper fits better in SciPost Physics Core rather than in SciPost Physics. We are convinced that our work represents a breakthrough in this field in identifying the peculiar information scrambling dynamics due to Hilbert space fragmentation tied to locality, which has not been discussed before. Furthermore, our analysis of this model establishes it as a much simpler and transparent model with Hilbert space fragmentation compared to previous fractonic and gauge theory models. As we show in the paper, it is very amenable to developing a deeper and transparent understanding of dynamical phenomena in such models. We have taken the first such step in our paper, highlighted by the remarkable agreement of theoretical and numerical results. We therefore think that this paper comfortably meets the SciPost Physics acceptance criteria: 1. Detail a groundbreaking theoretical discovery and 3. Open a new pathway in an existing research direction, with clear potential for multipronged follow-up work

A1. Comparison with the literature. The authors shall make an effort in explaining what makes their paper interesting to respect to what is already known in other systems displaying Hilbert space fragmentation.

We thank the referee for the question about the connection of our work to broader studies of Hilbert space fragmentation. In short, this paper is the first systematic study of the dynamics in the fully frustrated ladder that very nicely illustrates certain features of anomalous thermalization in fragmented models. As the Hilbert space fragmentation structure is particularly transparent in this model it is a good case to study to explore outstanding questions in the field.

In particular: (1) We show clearly that the model exhibits dynamical localization although the model is disorder-free. This is possible because one can choose the initial state to achieve localization lengths of the order of a lattice spacing to allow detailed numerical exploration of the phenomenon. (2) We show how this anomalous behavior arises from the fragment structure. For example, we show that large fragments, that are transparently nonintegrable, thermalize in a conventional manner leading to asymptotically random states. We also show that short length scale fragments have a very simple spectrum leading to Rabi oscillations in various dynamical signatures. Given these insights one may straightforwardly interpret the dynamics of the full model - both the entanglement dynamics and the operator spreading. The latter shows particularly rich features of short length scale oscillations, an emergent localization length and a long length scale thermalizing tail that is exponentially suppressed for larger system sizes. These are general features in any fragmented model tied to spatial locality. However in other models the fragment structure is more complex and interpretation of numerics more challenging.

We recognize that the introduction did not optimally spell out the open questions in the field and the accomplishments of this work in relation to those problems. We have therefore re-written the introduction to more clearly detail these two points. In doing so we have moved most of the details of the model we do study to the section immediately after the introduction.

A2. Numerics. Why is the TEBD simulation performed only for the entanglement entropy dynamics for α=0.5? Is there a technical reason? The authors shall try to improve the numerics, also using TEBD for larges system size when possible.

The TEBD simulations suffer from rapid entanglement growth, making late times eventually unaccessible. In our parameters, this concerns especially the case of large α and the computations of OTOCs (see also Kévin Hémery, Frank Pollmann, and David J. Luitz Phys. Rev. B 100, 104303 (2019)). Therefore, we restrict the use of TEBD to the regime of short times for the entanglement production curve in Fig. 3, where the results are converged with bond dimension chi=512. For larger alpha, as shown in Fig. 4, the system is much more entangled and therefore it is hard to get converged TEBD results in this regime. We would like to stress that in this model, as we discuss, there is a finite emergent localization length, which governs the physics. This length scale is much smaller than our system size in much of the parameter regime we consider and there is no need to consider larger systems here. For large alpha this is not true, but our analytical understanding established by rigorous comparison to numerics at smaller alpha extends to this challenging regime.

A3. The square commutator is studied only for the ^Sz operator. In the case of chaotic dynamics, the choice of the operator is inessential. However, the model under analysis is a strongly non-ergodic system. In this case, one expects relevant differences, especially if the operator is associated with the conservation law enforcing the fragmentation. It is claimed that the qualitative behaviour of the results is not affected by this choice. The authors shall show at least an example to corroborate this claim, even in the appendix.

The long time, long distance dynamics is determined by the local conservation laws and thus insensitive to the special choice of the operators. To illustrate this further, we have added an additional OTOC computation for local Sx/Sz operators in the appendix, showing the same phenomenology as for the Sz operator discussed in detail in the main text.

A4. The authors say that oscillations (of entanglement and the square-commutator) arise when the couplings J and J′ are commensurate. What happens when they are not?

The oscillations persist also for incommensurate couplings. We have removed this sentence.

A5. The effect of the lift of degeneracies is an interesting point discussed in the paper. The authors shall comment more on this issue. For instance, is it possible to estimate the saturation point value of the entanglement entropy? How would things change with different degeneracy breaking terms? How would things change with a single impurity? Is fragmentation always unstable?

Since additionally to the local conservation laws one has also total spin conservation, one obtains a saturation value below the Page value. However, due to the oscillatory behavior of the entanglement entropy, and the sensitivity to finite size effects, it is not possible to get very accurate estimates of the saturation value in the regime of small delta. The general behavior of the system is insensitive to single impurities, i.e they may break local conservation laws and thus affect the short distance behavior, but the behavior at long distances is not affected , as long most of the local conserved quantities stay preserved. Another interesting direction is the breaking of degeneracies by local magnetic fields. In that case, even if the local conservation laws are broken, one has instead so called local dynamical symmetries (Buca et al 2020,arXiv:2008.11166). They may lead to persistent oscillations in observables which have some overlap with these dynamical symmetries. However, their frequency is determined by the local magnetic field and thus of different kind as the oscillations observed in our case. We have added this discussion in our paper.

B1. Below figure 8, "the sum of the contributions Cl(i,t) of sectors with increasing lengths ..." is introduced. While the meaning of the sentence is quite intuitive, this notation has never been defined. It would be worth to state more clearly the quantities of interests.

We changed the order of the sentences to state this quantity more clearly.

B2. If the scope of Fig.7 is to show the scaling Cα(i>1,t)∝αi−2, why the authors did not plot directly that Cα(i>1,t)/αi−2?

We have now added an inset in the plot.

B3. The authors say "the maximal entropy scales linearly with the localization length for small lα". The inset of Fig.4a shows a collapse for small lα, which is not exactly linear (probably because of the small system sizes L). Is there another plot showing this?

Figure 12 shows the scaling at small system sizes and also the derivation of the leading order. We have added a sentence to refer to this appendix.

B4. Fig.11: missing x-axis label.

We have fixed it.

B5. Figs.6-10: missing system size N in the caption.

We have fixed it.

B6. The colours and lines of Fig.7 are confused. The authors shall try to improve the plot.

We have fixed it.

B7. The double point at the end of page 7.

We have fixed it.

---

## Round 4 · Referee Report · Anonymous (Referee 3) · 2021-8-30

Report

The authors have successfully addressed all of my questions and comments. I recommend publication in SciPost Physics.

---

## Round 4 · Referee Report · Anonymous (Referee 4) · 2021-9-2

Report

The authors have addressed some of my comments and the quality of the manuscript has significantly improved. However, they did not provide enough evidence to qualify the paper as a "breakthrough" in the field. Therefore, I still do not think that the paper meets the criteria of novelty required by SciPost Physics, but can I now recommend its publication in SciPost Physics Core.

---

## Round 4 · Author Response

We have replied to both reports directly

---

## Round 4 · List of Changes

- re-written the entire introduction
- added a second OTOC computation in the appendix
- added a new paragraph discussing the effects of lifted degeneracies
- fixed minor typos
- minor improvements on the plots

---

## Editorial Decision

published